# Towards Sustainable Food Services in Hospitals: Expanding the Concept of 'Plate Waste' to 'Tray Waste'

**Nouf Sahal Alharbi \***[ID]**, Malak Yahia Qattan**[ID] **and Jawaher Haji Alhaji**[ID]

Department of Health Sciences, College of Applied Studies and Community Service, King Saud University, Riyadh 11451, Saudi Arabia; mqattan@ksu.edu.sa (M.Y.Q.); jalhejjy@ksu.edu.sa (J.H.A.)
* Correspondence: noufsahal@ksu.edu.sa; Tel.: +96-65-0415-1280

**Abstract:** Early debates on the sustainability of food-plating systems in hospitals have concentrated mostly on plate waste food served, but not eaten. This study aims to address the need for more comprehensive studies on sustainable food services systems by expanding the concept of plate waste, to that of tray waste (organic and inorganic materials), through a case study of a hospital in Saudi Arabia. Tray waste arising at the ward level was audited for three weeks, covering 939 meals. It was found that, on average, each patient threw away 0.41, 0.30, 0.12, and 0.02 kg of food, plastic, paper, and metal, respectively, each day. All this equated to 4831 tons of food, 3535 tons of plastic, 1414 tons of paper, and 235 tons of metal each year at hospitals across Saudi Arabia. As all of this waste ends up in landfills, without any form of recycling, this study proposes the need for a more comprehensive, political approach that unites all food system stakeholders around a shared vision of responsible consumption and sustainable development.

**Keywords:** sustainability; food production and consumption; sustainable food systems; sustainable menu; food catering practices in the public sector

## 1. Introduction

Over the last few years, a large number of international organizations have recognized the economic and environmental impact of the waste generated by food systems [1,2]. According to the Food and Agricultural Organization (FAO), 1.3 billion tons of food are wasted every year, which costs around USD 936 billion [3–5]. At the international environmental level, it has been reported that food waste accounts for a portion of global carbon emissions, equivalent to that of a medium-sized country [6].

Within the various food sectors, hospital food waste has been estimated as being two to three times higher than other sectors, such as restaurants, work places, and schools [7]. Moreover, hospital food service waste can contribute to as much as half of the total waste generated in a ward [8,9]. Actually, from an economic and environmental perspective, in places, like the UK, Portugal, Brazil, and Saudi Arabia, the estimated hospital food waste costs ranged from USD 90,960 to USD 342,449 per year, while the average emission of $CO_2$ was estimated as 1.8 kg per patient, per day in Portugal [10–13]. As a consequence of this economic and environmental drain, early debates on the food industry and its sustainability have mostly concentrated on waste elimination and recycling, which were seen as critical strategies for creating a food system that promotes environmentally friendly practices. This is the main objective of the United Nations Sustainable Development Goals (UNSDGs) [14].

Previous international studies have shown that food service waste is mostly generated from production, cooking, and, lastly, at the point of the serving stage or plate waste [1,7,15]. Several studies carried out in the last decade have addressed plate waste by trying to quantify the amount of food waste arising from meal delivery services at hospitals [12,16–19]. However, none of these studies have

attempted to quantify both the food and its combined solid waste. In order to fill the gap, this paper sets out to extend the concept of plate waste to also include that of tray waste by analyzing, for the first time, the food and solid wastes arising within the hospital.

In order to do this, the paper focuses on a general hospital catering system in Riyadh, a city in Saudi Arabia, which is deeply committed to reviewing the status of the UN sustainable development goals and the country's alignment with Vision 2030. The Saudi government is aiming to achieve environmental sustainability, by preserving the natural resources and increasing the efficiency of waste management [20]. In this respect, the selected hospital in Riyadh, the country's capital city, provides an excellent research context to explore the extent to which the sustainability objectives were being translated into practice. Our study aims to provide new insights into the multiple diminutions of a sustainable food catering system by asking the following research questions: (1) What are the types and quantities of waste resulting from the catering services in hospitals? (2) What kind of sustainable measures do the food catering systems provide?

## 2. Materials and Methods

### 2.1. Case Study Description

The contemporary study was carried out at one of the biggest governmental tertiary hospitals in Riyadh city with 1200 beds, serving the various medical departments. For confidentiality purposes, the hospital has not been identified. The catering food system in the hospital offers 20 menu categories, aimed at meeting the nutritional requirements of the patients, according to their different health status. For example, apart from the normal diet menu, there are customized menus for patients with diabetes, renal disease, and other health conditions. Under each menu category, there are seven menus with a variety of food options for each day of the week, according to patient preferences. For example, in addition to the patient's selection of a cold or hot drink, breakfast also consists of packaged pita or toast bread and a choice from an array of main dishes, including beans, corn flakes, lentils or eggs, and two pieces of cheddar or cream cheese, honey and jam. Lunch and dinner include either a portion of rice or pasta with chicken, meat or fish, and mixed vegetables as the main course. Table 1 provides an example of a Sunday menu with the amounts of each meal for the diet of a typical patient. Every day, the menu is circulated to all in-patients between 1:30 and 2 p.m. for them to order their food for the next day. Using a computerized system, based on the list of patients according to their ward, room and diet requirements, the three meals are freshly cooked at a central kitchen in the hospital, and plated according to the patient's requirements. The patients are offered three main meals a day—breakfast at 6:30 a.m., lunch at 11 a.m., and dinner at 5:45 p.m. In addition to this, there are three refreshments snacks at 9 a.m., 1:30 p.m. and 5:45 p.m.

In the hospital, in some cases, patients can choose to share a room, or stay in one by themselves. However, patients with infectious diseases, or those who lack proper immunity are usually isolated. The main course for all isolated patients is served on plates made out of foil, and placed on cardboard trays. On the other hand, the meals for the non-isolated patients are served on a ceramic plate, placed on a reusable plastic tray. For all the patients (isolated or non-isolated), water, dairy drinks, sweets, fruits, bread and salads are served in plastic containers or packaging. Hot drinks are served in paper cups, and soup in foil plates, Figure 1. The meal is placed on a tray covered by a sheet of paper along with the paper menu. In addition, all the cutlery and cups are made from plastic or paper. The trays are transported to the wards in trolleys, and served to the patients at scheduled times.

After the meal, the trays are collected and transported back to the central kitchen, and all the tray waste is, at first, placed together, without any form of sorting, into waste bins (NAPCO Sanita, G.B.70 G SASO BIO, high-density polyethylene) and deposited into 800 L containers, located in the basement. They are then transported twice a day to the hospital's waste depot. From there, the waste is collected by a special private company, without any form of recycling procedures, it is then carried directly to the landfill sites.

**Table 1.** Inpatient normal diet Sunday menu (2800–3000 kcal).

| | Breakfast | Lunch | Diner |
|---|---|---|---|
| Water | 600 mL | 600 mL | 600 mL |
| Diary drink * | 200 mL Milk or Butter milk or 170 mL Yogurt | 200 mL Butter milk or 170 mL Yogurt | 200 mL Butter milk or 170 mL Yogurt |
| Hot drink * | 200 mL Tea or Coffee | 200 mL Tea or Coffee | 200 mL Tea or Coffee |
| Sweet and fruits * | 30 gm Honey or Jam | An Orange or 150 gm Pineapple or 150 gm Custard | A Banana or 150 gm Cream Caramel |
| Bread * | 125 gm (white or brown): Toast Bread or Pita Bread or Bun Bread | 125 gm (white or brown): Toast Bread or Pita Bread | 125 gm (white or brown): Toast Bread or Pita Bread |
| Main course * | Main course *: 50 gm Corn Flakes or 170 gm Lentil or 50 Shakshuka (scrambled egg with tomato) Cheese *: 50 gm Slice Cheese or Cream Cheese | Starches *: 200 gm Rice or 200 gm Pasta Meat *: 150 gm Grilled Chicken or 150 gm Grilled Fish or 100 gm Grilled Meat * Vegetables *: 150 gm Mixed vegetables or Cauliflower with Carrot sauté | Main course *: 150 gm Grilled Chicken + 200 gm Biryani Rice or 100 gm Grilled Lamb + 200 gm Biryani Rice or 100 gm Grilled Fish + 200 gm Biryani Rice or 2 Pieces Tuna Club Sandwich Vegetables *: 150 gm Cooked Bean or 150 gm Cooked Zucchini |
| Soup * | NA | 150 gm Mushroom Soup or Barley soup | 150 gm Mushroom Soup or Vermicelli Soup |
| Salad * | NA | 150 gm Green Salad or 150 gm Mixed Salad | 150 gm Green Salad or 150 gm Coleslaw Salad |

* based on patient preferences.

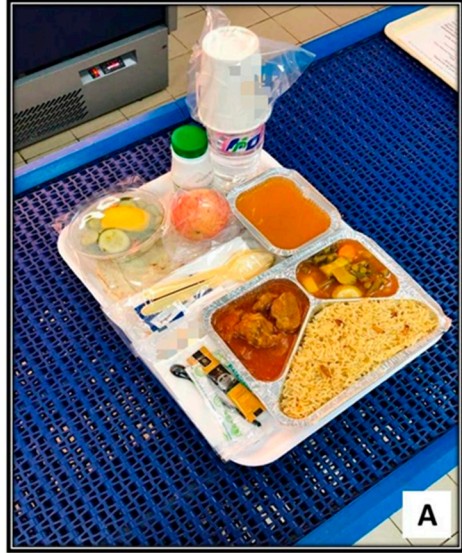 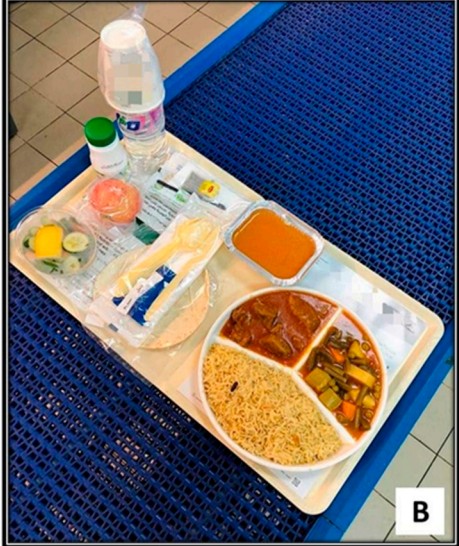

**Figure 1.** (**A**) A typical dinner for an isolated patient, including rice, meat, vegetables, soup, an apple and salad. (**B**) A meal for a non-isolated patient.

## 2.2. Waste Audit: Examination and Categorization

The waste was audited during the period of 15 September 15 to 6 October 2019 in eight wards, and consisted of 939 trays for the main meals. In this study, we included only the patients with solid diets, who represented approximately 89% of the total number of patients admitted to the hospital [21]. Patients with tube feeding, liquid diets and supplements were excluded. The data collection was carried out in two stages. Before the meal was served, all data about all the tray components and the weight of the meals by the ward name and bed type were obtained from the electronic food services system. In order to increase the data accuracy, a random sample from each meal was weighed on a digital scale (MOTEXT weight scale ML 30 N). At this first stage, we recorded the data on an Excel sheet, Figure 2. At the second stage, the overall food waste from each ward for the different medical

departments was first sorted daily and weighed separately; then empty packaging and other tray waste were sorted and weighed separately, according to the waste type as follows: plastic, paper, and metal. The waste containers (NAPCO Sanita, G.B.70 G SASO BIO, high-density polyethylene) with tags were set aside in a specified area in the main kitchen to be weighed. All of the waste was weighed three times a day—the morning sample included waste arising from breakfast, the afternoon sample included waste arising from lunch, and the evening sample included waste arising from dinner. Since food is consumed, unlike the other waste made of plastic, paper and metal, we determined the plate waste by dividing the amount of food waste by the amount of food served, using the following equation:

$$\text{Plate waste \%} = \text{Food waste/Food served} \times 100$$

| | non-ISO | ISO | Patient No. | Ward | Meal | Day | Date | | |
|---|---|---|---|---|---|---|---|---|---|
| | 14 | 12 | 26 | ADULT ACUTE CARE | Lunch | Friday | 11/10/19 | | |

| # | Water | Dairy Pro. | Drink | Sweet | Fruit | Bread | Starches | Meat | Vegetables | Soup | Salad |
|---|---|---|---|---|---|---|---|---|---|---|---|
| | | | | | | | PATIENT MEALS | | | | |
| 1 | 1 (600)ML | L (200) ML | T | | B 150 G | 125 GM | R (200) GM | C (150) GM | 150 G | 150 G | 150 G |
| 2 | 1 (600)ML | L (200) ML | T | | B 150 G | 125 GM | R (200) GM | C (150) GM | 150 G | 150 G | 150 G |
| 3 | 1 (600)ML | Y (200) ML | T | | B 150 G | 125 GM | R (200) GM | C (150) GM | 150 G | 150 G | 150 G |
| 4 | 1 (600)ML | L (200) ML | T | | B 150 G | 125 GM | R (200) GM | C (150) GM | 150 G | 150 G | 150 G |
| 5 | 1 (600)ML | L (200) ML | C | | B 150 G | 125 GM | R (200) GM | M (100) GM | 150 G | 150 G | 150 G |
| 6 | 1 (600)ML | L (200) ML | C | CC 150 | | 125 GM | R (200) GM | M (100) GM | 150 G | 150 G | 150 G |
| 7 | 1 (600)ML | Y (200) ML | C | CC 150 | | 125 GM | R (200) GM | C (150) GM | 150 G | 150 G | 150 G |
| 8 | 1 (600)ML | Y (200) ML | T | CC 150 | | 125 GM | R (200) GM | M (100) GM | 150 G | 150 G | 150 G |
| 9 | 1 (600)ML | L (200) ML | C | CC 150 | | 125 GM | R (200) GM | M (100) GM | 150 G | 150 G | 150 G |
| 10 | 1 (600)ML | L (200) ML | C | CC 150 | | 125 GM | R (200) GM | C (150) GM | 150 G | 150 G | 150 G |
| 11 | 1 (600)ML | L (200) ML | T | CC 150 | | 125 GM | R (200) GM | C (150) GM | 150 G | 150 G | 150 G |
| 12 | 1 (600)ML | L (200) ML | T | CC 150 | | 125 GM | R (200) GM | F (150) GM | 150 G | 150 G | 150 G |
| 13 | 1 (600)ML | L (200) ML | C | CC 150 | | 125 GM | R (200) GM | F (150) GM | 150 G | 150 G | 150 G |
| 14 | 1 (600)ML | L (200) ML | T | CC 150 | | 125 GM | R (200) GM | C (150) GM | 150 G | 150 G | 150 G |
| 15 | 1 (600)ML | L (200) ML | C | CC 150 | | 125 GM | R (200) GM | C (150) GM | 150 G | 150 G | 150 G |
| 16 | 1 (600)ML | L (100) ML | T | | A 150 G | 125 GM | R (125) GM | C (120) GM | 150 G | 100 G | 150 G |
| 17 | 1 (600)ML | L (100) ML | C | | A 150 G | 125 GM | R (125) GM | M (100) GM | 150 G | 100 G | 150 G |
| 18 | 1 (600)ML | L (100) ML | C | | A 150 G | 125 GM | R (125) GM | F (100) GM | 150 G | 100 G | 150 G |
| 19 | 1 (600)ML | Y (100) ML | C | | O 150 G | 125 GM | R (125) GM | F (100) GM | 150 G | 100 G | 150 G |
| 20 | 1 (600)ML | L (100) ML | T | | O 150 G | 125 GM | R (125) GM | M (100) GM | 150 G | 100 G | 150 G |
| 21 | 1 (600)ML | L (100) ML | T | | A 150 G | 125 GM | R (125) GM | C (120) GM | 150 G | 100 G | 150 G |
| 22 | 1 (600)ML | Y (100) ML | T | | O 150 G | 125 GM | R (125) GM | M (100) GM | 150 G | 100 G | 150 G |
| 23 | 1 (600)ML | L (100) ML | C | | O 150 G | 125 GM | R (125) GM | M (100) GM | 150 G | 100 G | 150 G |
| 24 | 1 (600)ML | L (100) ML | T | | O 150 G | 125 GM | R (125) GM | C (120) GM | 150 G | 100 G | 150 G |
| 25 | 1 (600)ML | L (100) ML | T | | A 150 G | 125 GM | R (125) GM | C (120) GM | 150 G | 100 G | 150 G |
| 26 | 1 (600)ML | L (100) ML | T | | A 150 G | 125 GM | R (125) GM | M (100) GM | 150 G | 100 G | 150 G |
| Total | 15.6 | 4.3 | | 1.5 | 2.4 | 3.25 | 4.375 | 3.23 | 4.5 | 3.23 | 3.9 |

| Shortcut Detalis ( Drink ) | |
|---|---|
| Tea | T |
| Coffee | C |
| Juice | J |

| Shortcut Details ( Dairy ) | |
|---|---|
| Milk | M |
| Laban | L |
| Yogurt | Y |

| Shortcut Details ( MAET ) | |
|---|---|
| Maet | M |
| Chicken | C |
| Fish | F |

| Skortcut Detalis ( Starches ) | |
|---|---|
| Rice | R |
| Macaroni | M |

| Shortcut Details ( Sweet ) | |
|---|---|
| Jello | J |
| Custard | CU |
| Cream caramel | CC |

| Shortcut Details ( Fruit ) | |
|---|---|
| Apple | A |
| Orange | O |
| Fruit salad | FS |
| Pineapple | P |
| Grape | G |
| Banana | B |

**Figure 2.** An example of the data collection sheet for lunch meal.

Untouched main meals were individually counted and their weight was included in the waste. Protective clothes were worn during the categorization and quantification of the waste.

*2.3. Statistical Analysis*

Data entry and analysis were conducted using the IBM Statistical Package for Social Science (SPSS) version 20.0 (Armonk, NY, USA). Statistical analysis procedures included a descriptive analysis of the total amount of each type of waste, and the means and the confidence intervals of each type for the three main meals per patient per day were computed separately. Finally, after verification that the data were normally distributed, we examined the association between tray waste and bed type using the *t*-test. The statistical significance level was assumed for all estimations as *p* value ≤ 0.05. Values are presented as means and confidence intervals.

**3. The Results**

The average tray waste of the food, paper, plastic, and metal were 0.41, 0.30, 0.12, and 0.02 kg per patient per day, respectively. A comparison of the tray waste showed that the paper and metal waste levels were significantly higher among isolated patients—0.21 vs. 0.08, and 0.034 vs. 0.016–kg per patient per day, respectively, with no statistical significance for other tray waste types, Figure 3.

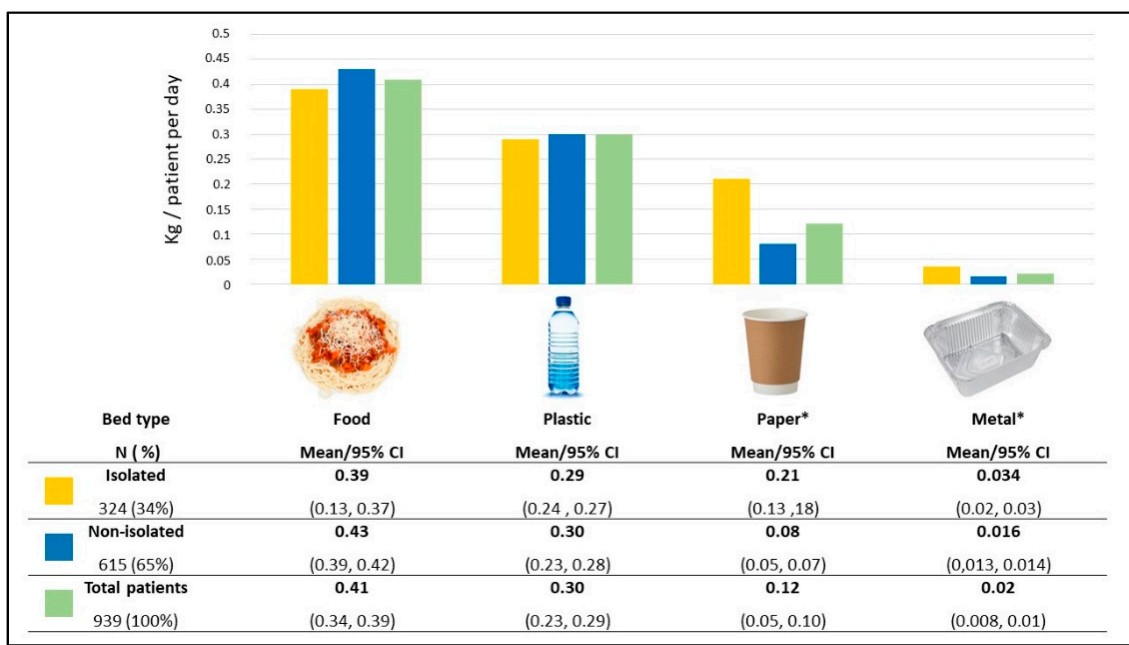

**Figure 3.** A comparison for tray waste by bed type for all meals per patient per day.

With regard to the plate waste across the main meals, although statistical relevance was not found, the lowest plate waste recorded for lunch was 15.5%, while the highest plate waste was 22% for dinner. On average, each patient threw away 412 g of food each day, representing 18.2% of the total food served. Waste, in accordance with the different main meals, is presented in Table 2 and Figure 4.

**Table 2.** Summarized mean weight tray waste (kg) per patient per day in accordance with meal time.

| Waste | Breakfast | Lunch | Dinner |
|---------|-----------|--------|--------|
| Food | 0.08 | 0.15 | 0.16 |
| Paper | 0.042 | 0.051 | 0.039 |
| Plastic | 0.0974 | 0.0991 | 0.097 |
| Foil | 0.0063 | 0.01 | 0.0072 |

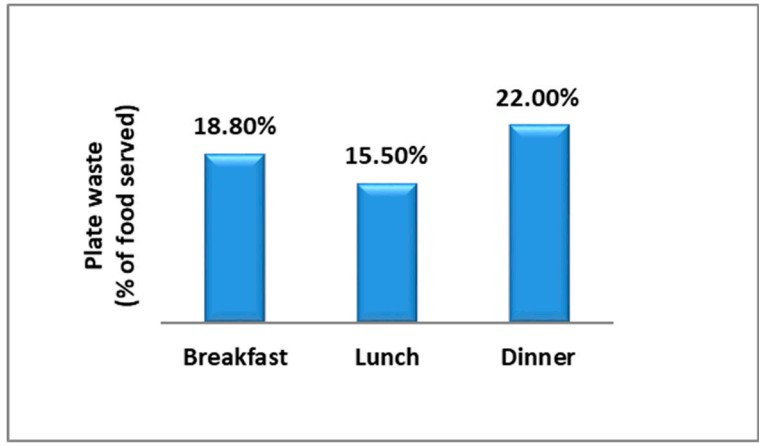

**Figure 4.** Plate waste per patient across the main meals.

## 4. Discussion

To the best of our knowledge, this was the first study that yielded a comprehensive picture about the extent of sustainability in food catering services, by using the tray as an assessment unit. In this study, we audited the tray waste (939 trays in all) at the ward level for three weeks, using a general hospital as a case study. Our results revealed that the overall food waste was 412 g per patient per day, and this figure was nearly similar to the average net of all inorganic wastes (plastic, paper, and metal) of 441 g per patient per day. However, according to the patient's bed type, this study found that the total amount of inorganic waste exceeded that of the food waste, where the average estimated inorganic waste was 534 g per patient per day. This figure is statistically higher by 34% than the waste generated from the patients who stayed in shared rooms.

The plate waste values of previous international studies conducted in the UK, Brazil, Portugal, the Netherlands and Australia, applying a similar assessment method in the context of hospital food services, ranged from 29% to 42%. Compared to our study, the results showed a lower value of plate waste at 18% [12,16–19]. In addition, the amount of plate waste arising from this Saudi case study was almost 40% lower (412 g) than that estimated in another study conducted in a general hospital in Portugal, where it was 953 g. This was so for these two studies, the food services department applied the same food serving system of "plating, not bulk". However, the dissimilarity in the results of the plate waste might be due to the differences in the food services systems. In the Saudi study, the meals were freshly cooked every day and the plating was according to the patients' preferences, while in the Portugal study, the food preparation was based on the cook-chill method, and the patients had limited options when choosing from the menu [12].

According to the latest national official statistics in Saudi Arabia in 2018, there were 284 government hospitals with a total of 43,690 beds. During that year, it was estimated by the Saudi Ministry of Health that, for these hospitals, a total of 35 million solid meals was provided [22]. Taking into consideration these numbers and our findings, this equates food tray waste in governmental hospitals in the country amounting to the discarding of about 4831 tons of food, 3535 tons of plastic, 1414 tons of paper, and 235 tons of metal each year. Thus, these indicators represent both a challenge to, and an opportunity for, the Saudi government.

From a sustainability point of view, by comparing retrospectively our food waste results with those from previous studies, it seems that the Kingdom of Saudi Arabia has made a remarkable move in achieving food security—one of the sustainable development goals [10,14]. However, on the other hand, there are still opportunities for stakeholders to meet the challenges of responsible consumption and production, which is another main sustainable development goal [14]. From a political point of view, in order to ensure that the Saudi government minimizes the carbon emissions associated with healthcare waste landfill, they can consider recycling to help reduce the depletion of plastic, paper and metal. Furthermore, the sustainable handling of food waste can return nutrients to the soil [23]. To achieve this, our study emphasizes the importance of developing a more integrated strategy to manage the waste—organic and inorganic—generated by the food systems in Saudi Arabia. This can be achieved by creating a legislative organization that mobilizes and unifies the practices of all the actors in the food industry, in order to create a shared vision for sustainable development in the country.

## 5. Conclusions

This was the first study that explored the extent of food service sustainability practices in Saudi hospitals. All the estimated tons of food, paper, plastic and metal transformed into waste equate to an environmental impact and economic losses. Indeed, the figures presented in our study highlight the opportunity for financial and environmental savings that can accrue to the Saudi health system by tackling this challenge. However, despite the contribution of this paper, it was challenging to compare our results with those obtained in other studies, due to the novelty of the research method that was characterized by our adopting of the tray as a new assessment unit. The plate waste results drew attention to the analysis of all waste, resulting from the different food preparation stages. Therefore,

it would be interesting to gain insights into the reasons for the amount of plate waste generated in Saudi hospitals by conducting an in-depth analysis that included the perspectives of both the food service staff and the patients. In addition, further research is needed to evaluate the long-term environmental costs to society, and the possible measures to be adopted for cost-saving with regards to the food service budget. Other areas for investigation could also include water and energy usage and carbon emissions.

**Author Contributions:** N.S.A.; Conceptualization; Project administration; Methodology, reviewing and editing the final reaft M.Y.Q.; Data collection; Writing first draft, J.H.A.; Data collection; Validation; Reviewing and editing. All authors have read and agreed to the published version of the manuscript.

**Funding:** This research received no external funding.

**Acknowledgments:** The authors extend their appreciation to the Research Center for Humanity, Deanship of Scientific Research at King Saud University for funding this Research Group No. (HRGP-1-19-05).

**Conflicts of Interest:** The authors declare that they have no competing interests.

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
