# Peer review of "Towards Sustainable Food Services in Hospitals: Expanding the Concept of ‘Plate Waste’ to ‘Tray Waste’"

_sustainability, doi:10.3390/su12176872_

Round 1

Reviewer 1 Report

The article "Towards sustainable Food Services in Hospitals:Expanding the concept of Plate to tray waste" deals with an interesting topic, but I think the approach needs improvement.

First of all, even the study is interesting and the results are valuable for the industry, the information is poorly presented.

Introduction - is too short, can be improved; the information is too generally presented.

Line 38 - CO2 instead of Co2

Line 54 - Vision 2030 programme should be shortly described in order to discover the correlation with this study.

Is missing the objective of the study, in the last part of the introduction.

Materials and methods - 2.1.  part is well described

Lines 76-77: Is the meal scheduale correct? because it is written the same hour for  dinner and the last refreshment snack - 5:45 pm.

Maybe it will be more interesting if authors will describe in detail how they obtained the results presented (how they sort the waste, how they handled the waste organization etc.).

2.3. part - must be improved with more graphs and information presented could be more highlghted.

I think that data presented can be organized in another way (tables, graphs) to be easier to follow.

Discussion part - is well described, pertinent comparisons with the results from other specialized articles are observed.

Conclusions - can be improved.

Reviewer 2 Report

Dear Authors,

The article presented for review is very interesting and deals with an important topic of food waste and generated other waste in catering, in this case, hospital catering.

Please note and address the following comments:

Abstract

In the abstract, authors wrote ‘All this equated to 4, 831 tons of food, 3,535 tons of plastic, 1,414 tons of paper and 235 tons of metal each year at hospitals across Saudi Arabia’.  Is estimation based on the results from this manuscript or results from other analyses? If it results from other analysis these counts shouldn't be presented in the abstract

Figure 3 is connected with Table 3, although this Figure is interesting in my opinion authors should add another line e.g. average waste per patient per day to Table 3. It will be clearer/simpler for readers.

Conclusion

Authors wrote that would be interesting to gain insights into the reasons for the amount of plate waste generated in Saudi hospitals by conducting an in-depth analysis that included the perspectives of both the food service staff and the patients. Why didn’t they try to answer this question themselves? Their observation from the hospital would be helpful for the next research.

In my opinion, the manuscript requires minor language correction

The manuscript presented for review is very interesting. Despite my comments, I recommend the article for publication in the Sustainability journal. In my opinion, the paper is important for the world scientific society. The topic of the article is relevant and interesting.

Reviewer

Author Response

Please see the attchment

Reviewer 3 Report

Thank you for the opportunity to review this paper. This study is very important in systematically calculating the vast amounts of waste generated in healthcare foodservice and in considering packaging and other forms of waste beyond food. 

Some comments:

It might be a stretch to say "which is one of the few countries deeply committed to reviewing the status of 54 the UN sustainable development goals and their alignment with Vision 2030" Maybe "one country deeply" would be less ethnocentric.

Line 71--"normal patient" is pejorative to others. "typical patient" would be more appropriate

More attention in the discussion is needed to the benefits of individual packaging from a food safety perspective and comparative details on labor and water (especially important in Saudi Arabia) required for washing if reusable dishes were used. While waste is bad, foodborne illness or other costs like labor and water usage need to be considered. 

Round 2

Reviewer 1 Report

În my opinion, the paper can be published in the present form. It is obvious that authors improved their manuscript.